# COUPLED ENSEMBLES OF NEURAL NETWORKS

**Anuvabh Dutt, Denis Pellerin & Georges Quénot**
Univ. Grenoble Alpes, CNRS, Grenoble-INP, LIG, Gipsa-Lab, F-38000 Grenoble France
`{firstname.lastname}@univ-grenoble-alpes.fr`

## ABSTRACT

We present coupled ensembles of neural networks, which is a reconfiguration of existing neural network models into parallel branches. We empirically show that this modification leads to results on CIFAR and SVHN that are competitive to state of the art, with a greatly reduced parameter count. Additionally, for a fixed parameter, or a training time budget coupled ensembles are significantly better than single branch models. Preliminary results on ImageNet are also promising. Code for the experiments can be found at: `https://github.com/vabh/coupled_ensembles`

## 1 INTRODUCTION

The design of early convolutional architectures (CNN) involved choices of hyper-parameters such as: filter size, number of filters at each layer, and padding (LeCun et al., 1998; Krizhevsky et al., 2012). Since the introduction of the VGGNet (Simonyan & Zisserman, 2014), ResNet (He et al., 2016) and DenseNet (Huang et al., 2017), the design has moved towards following a template: filter size of $3 \times 3$ and $N$ features maps, down-sample to half the input resolution *only* by the use of either `maxpool` or strided convolutions (Springenberg et al., 2015), doubling the number the feature maps following each down-sampling operation, and "skip-connections" between non-contiguous layers.

Our work extends this template by adding another element, which we refer to as "coupled ensembling". In this set-up, the network is decomposed into several branches, each branch being functionally similar to a complete CNN. We show that given a parameter budget, it is better to have the parameters split among branches rather than having a single branch (which is the case for all current networks). The activations of the parallel branches are combined by taking the arithmetic mean of the individual log-probabilities. Combining these elements, we significantly match and improve the performance of convolutional networks on CIFAR and SVHN datasets, with a heavily reduced parameter count.

## 2 COUPLED ENSEMBLES

For the following discussion, we define some terms:

- Branch: the proposed model comprises several branches. The number of branches is denoted by $e$. Each branch takes as input a data point and produces a score vector corresponding to the target classes. Current design of CNNs are referred to as single-branch, having $e = 1$.
- Element block: the model architecture used to form a branch. In our experiments, we use DenseNet-BC and ResNet with pre-activation as element blocks.
- Fuse Layer: the operation used to combine each of the parallel branches which make up our model. In our experiments, each of the branches are combined by taking the mean of each of their individual log probabilities over the target classes.

We consider neural network models which output a score vector of the same dimension as the number of target classes. This is usually implemented as a linear layer and referred to as a fully connected (FC) layer. The differences among current neural network models is related to what is present before the last FC layer. We are agnostic to this internal setup (however complex it may or may not be) because the resulting "element block" always takes an image as input and produces a vector of $N$ values as output, parametrized by a tensor $W$.

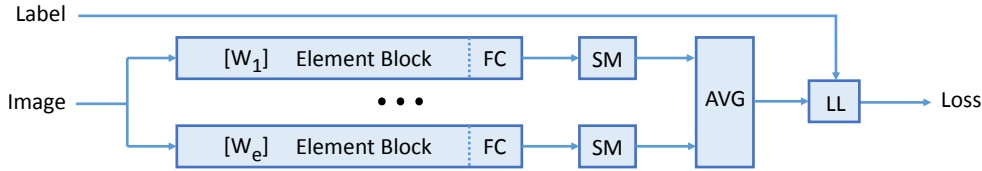

Figure 1: Coupled ensembles with mean of LogSoftMax for the fuse layer operation.

Table 1: Classification error comparison with the state of the art. The third and fourth groups of rows compare coupled ensembles with single branch models for the same parameter budget.

| Architecture | C10+ | C100+ | SVHN | #Params |
|---|---|---|---|---|
| ResNet pre-act. $L = 164\ k = 64$ (He et al., 2016) | 5.46 | 24.33 | - | 1.7M |
| ResNet pre-act. $L = 1001\ k = 64$ | 4.92 | 22.71 | - | 10.2M |
| ResNet pre-activation $L = 65\ k = 64\ e = 2$ | 5.26 | 23.24 | - | 1.4M |
| ResNet pre-activation $L = 164\ k = 64\ e = 4$ | 3.96 | 18.84 | - | 6.8M |
| DenseNet-BC $L = 100\ k = 12\ e = 1$ | 4.77 | 22.87 | 1.79 | 0.8M |
| DenseNet-BC $L = 112\ k = 16\ e = 1$ | 4.47 | 20.73 | 1.83 | 1.7M |
| DenseNet-BC $L = 130\ k = 20\ e = 1$ | 3.86 | 19.62 | 1.84 | 3.4M |
| DenseNet-BC $L = 160\ k = 24\ e = 1$ | 3.74 | 18.43 | 1.88 | 6.9M |
| DenseNet-BC $L = 166\ k = 32\ e = 1$ | 3.68 | 17.68 | 1.88 | 13.0M |
| DenseNet-BC $L = 190\ k = 40\ e = 1$ | 3.75 | 17.22 | 1.79 | 25.8M |
| DenseNet-BC $L = 82\ k = 8\ e = 3$ | 4.30 | 21.25 | 1.66 | 0.8M |
| DenseNet-BC $L = 82\ k = 10\ e = 4$ | 3.78 | 19.92 | 1.62 | 1.6M |
| DenseNet-BC $L = 88\ k = 14\ e = 4$ | 3.57 | 17.68 | 1.55 | 3.5M |
| DenseNet-BC $L = 88\ k = 20\ e = 4$ | 3.18 | 16.79 | 1.57 | 7.0M |
| DenseNet-BC $L = 94\ k = 26\ e = 4$ | 3.01 | 16.24 | **1.50** | 13.0M |
| DenseNet-BC $L = 106\ k = 33\ e = 4$ | 2.99 | **15.68** | 1.53 | 25.1M |

In our set-up, the model is composed of parallel branches and each branch produces a score vector for the target categories. The score vectors are combined through the "fuse layer" during training and the composite model produces a single prediction. We refer to this as coupled ensembles (Figure 1). No additional parameters are introduced as the fuse layer is an arithmetic operation without any learnable parameters.

## 3 EXPERIMENTS AND RESULTS

We use DenseNet Huang et al. (2017) as the element block. We train and evaluate on CI-FAR Krizhevsky & Hinton (2009), SVHN Netzer et al. (2011) and ILSVRC2012 Russakovsky et al. (2015) datasets. All yyperparameters for training set according to the description provided in the papers of the element blocks.

### 3.1 COMPARISON WITH SINGLE BRANCH MODELS

We fix the parameter budget and compare the top-1 error for single branch DenseNet and coupled ensembles with DenseNet element blocks. The results are in table 1. We see that for all parameter counts, coupled ensembles ($e > 1$) perform significantly better. DenseNet-L88-k20-e4 has error of 16.79% with 7M parameters, which is better than the 25M parameter single branch DenseNet-L190-k40-e1 model's 17.22%. Additionally, with 25M parameters, a coupled ensemble model achieves an error of 15.68%. In this case, coupled ensembles exceed the performance of single branch model with 7× less parameters, and for the same number of parameters perform better by 2%. The first four rows of table 1 show that using ResNet as the element block has similar results, with coupled ensembles performing significantly better.

Table 2: Coupled Ensembles of DenseNet-BC ($e = 4$) with different "fuse layer" combinations versus a single branch model. Performance is given as the top-1 error rate (mean±standard deviation for the individual branches) on the CIFAR-100 test set. Columns "$L$" and "$k$" denote the "element block" architecture, "$e$" is the number of branches. Column "Avg." indicates the type of "fuse layer" during training (section 2); Column "Individual" is error of each branch; Columns "FC" and "SM"(SoftMax) give the performance for "fuse layer" choices during inference.

| $L$ | $k$ | $e$ | Avg. | Individual | FC | SM | Params |
|---|---|---|---|---|---|---|---|
| 100 | 12 | 4 | LSM | 22.29±0.11 | **17.61** | 17.68 | 3.20M |
| 100 | 12 | 4 | none | 23.13±0.09 | 18.42 | 18.85 | 3.20M |
| 100 | 25 | 1 | n/a | 20.61 | n/a | n/a | 3.34M |
| 154 | 17 | 1 | n/a | 20.02 | n/a | n/a | 3.29M |

Table 3: Preliminary results on ImageNet with DenseNet element blocks.

| $L$ | $k$ | $e$ | Params. | Epochs | Train time (h) | Top-1 error |
|---|---|---|---|---|---|---|
| 161 | 32 | 1 | 14.1M | 90 | 162 | 31.21 |
| 121 | 30 | 2 | 14.1M | 90 | 225 | 29.41 |
| 121 | 30 | 2 | 14.1M | 64 | 160 | 29.83 |

### 3.2 COMPARISON WITH INDEPENDENTLY TRAINED MODELS

We next compare coupled ensembles to an ensemble of independently trained models. Row 4 of table 2 shows the results obtained by an ensemble 4 DenseNet-L100-k12 models, each of which where trained seperately. During inference, the models are combined by taking the mean of their predicted log-probabilities. We compare this with DenseNet-L100-k12-e4, where each branch is functionally equivalent to the individual single branch DenseNet models. First we see that error for coupled ensembles is $17.61\%$, which is lower than the ensemble of independent models' $18.42\%$.

In table 2, the column 'Individual' shows the error rate of branches. We observe that coupled ensembles' branches have an error rate of $22.29 \pm 0.11$. In contrast independent trainings of equivalent models obtains an error rate of $23.13 \pm 0.09$. This shows that the joint training in coupled ensembles aids each branch to perform better than when they are trained separately.

### 3.3 IMAGENET

Preliminary results on the ILSVRC-2012 validation set show that coupled ensembles have a lower error for a fixed parameter budget, as compared to a single branch model (DenseNet-L169-k32-e1) (table 3). We can see that results from the smaller scale datasets are carried over to the larger scale ILSVRC dataset. We also see that coupled ensembles perform better for a fixed training time (row 1 and 3 of table 3). Note that due to resource constraints, this was the strongest possible baseline that we could train. Training was done on down sampled $256\times256$ images instead of taking crops from the full sized images. Data augmentation consisted of random horizontal flips and random crops.

## 4 CONCLUSION

Coupled ensembles are a way of reconfiguring neural networks into 'element blocks', which resemble CNNs. This improves upon existing models especially for a fixed parameter budget, leading to competitive state of the art results. Additionally, the performances of the constituent element blocks are better than the case where the same element blocks are trained independently without coupling.

ACKNOWLEDGMENTS

This work has been partially supported by the LabEx PERSYVAL-Lab (ANR-11-LABX-0025-01). Some experiments presented in this paper were carried out using the Grid'5000 testbed, supported by a scientific interest group hosted by Inria and including CNRS, RENATER and several Universities as well as other organizations (see `https://www.grid5000.fr`)

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
