# OpenReview forum: "Coupled Ensembles of Neural Networks"
_ICLR.cc/2018/Workshop — Accept_

### Official Review · AnonReviewer1 · 2018-02-24
**Interesting observation for neural network ensemble**

**Rating:** 6
**Confidence:** 4

**Review:**

This paper introduces a meta neural network structure with multiple branches. Each of the branches outputs a softmax probability, which is them averaged over the branches as then final output the whole network. The authors empirically show that jointly training these branches yields better results than training the branches independently.

Overall, I think this is an interesting finding, as usually we would expect that independantly trained models may have more diversity, and thus lead to better ensemble performance. The results shown here seem to suggest that couple training is actually more effective. I'm curious to see how the model diversity obtained by the proposed method compared to training models independently, by perform an analytical experiments similar to that in ref [1] given below. I would also expect that more comparisons against indepent training on multple datasets.

Finally, the deeply fused network and fractalnet are two related network architectures, which should be discussed in the related work.


[1] Snapshot Ensemble: Train 1, Get M for Free. Huang et al, ICLR 2017
[2] FractalNet: Ultra-Deep Neural Networks without Residual, Larsson et al, ICLR 2017
[3] Deeply-Fused Nets. Wang et al, arXiv 2016

---

### Official Review · AnonReviewer3 · 2018-03-07
**Deep model with coupled branches**

**Rating:** 5
**Confidence:** 3

**Review:**

In this work, the authors propose to design a light-weight and accurate deep model, by coupling several branches at the softmax layer. Here are some comments:
1 The idea is not quite novel, since MobileNet, ShuffleNet are the similar kind of model type via group convolutions.
2 The literature review of light-weight model design is missing.

---

### Official Review · AnonReviewer2 · 2018-03-07
**Paper is at a test of concept stage and shows promising experimental results. It does not provide much discussion on why it works well.**

**Rating:** 6
**Confidence:** 3

**Review:**

Quality:

The paper proposes a neural network architecture that fuses multiple parallel neural network architectures into one architecture. This is achieved by branching of standard NN architectures, each taking an image as input and producing outputs to be combined in the last layer. The paper shows promising results on improvement in accuracy compared to architectures without branching even when ensemble uses smaller number of parameters and it also performs better compared to to the ensemble of independent NN outputs via averaging.

The paper does not provide much discussion regarding why it works well compared to the baseline. And the experiments on ImageNet data seems incomplete due to computing resource constraints. The paper does not touch up on ease of implementing this architecture using existing NN libraries. However, the paper is at a test-of-concept stage and provides good experimental results on non-trivial dataset.

Clarity:

The paper is written in simple terms and hence, easy to follow. There seems to be small typos:

- In section 3.2, "Row 4 of table 2 shows the results obtained by an ensemble 4 DenseNet-L100-k12 models, each of which where trained seperately." I think it should be row 2 of table 2 not row 4.

- "All yyperparameters" -> "All hyperparameters"

Significance: It can become a useful in various NN applications.

Originality: N/A. This reviewer is not knowledgeable enough to comment on the originality of the work.

---

### Decision · Program_Chairs · 2018-03-20
**ICLR 2018 Workshop Acceptance Decision**

**Decision:**

Accept

**Comment:**

Congratulations, your paper was accepted to the ICLR workshop.